# Molecular Ultrasound Imaging Depicts the Modulation of Tumor Angiogenesis by Acetylsalicylic Acid

**DOI:** 10.3390/ijms24087060

**Published:** 2023-04-11

**Authors:** Flurin Mueller-Diesing, Wiltrud Lederle, Anne Rix, Susanne Koletnik, Dennis Doleschel, Maximilian Snelting, Felix Gremse, Fabian Kiessling

**Affiliations:** Institute for Experimental Molecular Imaging, Helmholtz Institute for Biomedical Engineering, RWTH Aachen University, Forckenbeckstrasse 55, 52074 Aachen, Germany; fmueller@ukaachen.de (F.M.-D.);

**Keywords:** COX inhibitor, CEUS, molecular ultrasound, VEGFR-2, acetylsalicylic acid, breast cancer

## Abstract

Acetylsalicylic acid (ASA) is a well-established drug for heart attack and stroke prophylaxis. Furthermore, numerous studies have reported an anti-carcinogenic effect, but its exact mechanism is still unknown. Here, we applied VEGFR-2-targeted molecular ultrasound to explore a potential inhibitory effect of ASA on tumor angiogenesis in vivo. Daily ASA or placebo therapy was performed in a 4T1 tumor mouse model. During therapy, ultrasound scans were performed using nonspecific microbubbles (CEUS) to determine the relative intratumoral blood volume (rBV) and VEGFR-2-targeted microbubbles to assess angiogenesis. Finally, vessel density and VEGFR-2 expression were assessed histologically. CEUS indicated a decreasing rBV in both groups over time. VEGFR-2 expression increased in both groups up to Day 7. Towards Day 11, the binding of VEGFR-2-specific microbubbles further increased in controls, but significantly (*p* = 0.0015) decreased under ASA therapy (2.24 ± 0.46 au vs. 0.54 ± 0.55 au). Immunofluorescence showed a tendency towards lower vessel density under ASA and confirmed the result of molecular ultrasound. Molecular US demonstrated an inhibitory effect of ASA on VEGFR-2 expression accompanied by a tendency towards lower vessel density. Thus, this study suggests the inhibition of angiogenesis via VEGFR-2 downregulation as one of the anti-tumor effects of ASA.

## 1. Introduction

Ultrasound is routinely used in the clinic to diagnose and monitor tumors and their therapy responses [1,2,3]. In this context, contrast agents allow assessing functional and molecular changes. Microbubbles, composed of a shell of phospholipids, proteins, or polymers and a gas core, are applied as contrast agents. The properties of the contrast agents can be varied via both the composition of the shell and the gas filling. For example, more stable shells can yield greater microbubble longevity [4]. Similarly, better post-injection durability can be achieved with poorly soluble gases (e.g., sulfur hexafluoride or perfluorocarbons) [5]. The usual diameter of the microbubbles is between 1 and 4 µm [6]. If microbubbles are injected intravenously, they remain strictly intravascular due to their size and then provide an increased linear backscatter of ultrasound waves in blood vessels. Additionally, under the influence of ultrasound waves, non-linear responses are emitted from the microbubbles, enabling their differentiation from the surrounding tissues [4,7].

Modifications of the microbubble shell allow their binding to specific markers at cellular surfaces. In this way, changes can be assessed at the molecular level.

On preclinical microbubbles, targeting moieties are often conjugated using avidin and biotin. Here, the avidin is attached to the microbubble’s surface and binds the biotinylated targeting moiety (e.g., antibody), the latter binding to the cellular targets [4]. Since the microbubbles cannot pass through the vascular endothelium, molecules of the vascular endothelium often serve as targets, such as VEGFR-2 or α_v_β_3_ integrin [8].

Due to its prominent role in tumor angiogenesis and overexpression in tumors, VEGFR-2 is an established target structure for targeted microbubbles [4,6,9]. In previous studies, Palmowski et al. successfully demonstrated therapeutic effects of a matrix metalloproteinase inhibitor at the molecular level on squamous cell carcinoma xenografts in mice using VEGFR-2-targeted microbubbles [10]. Baetke et al. showed that VEGFR-2-specific microbubbles are more sensitive than nonspecific microbubbles in assessing the therapy response of squamous cell carcinoma during anti-angiogenic therapy [11].

Acetylsalicylic acid (ASA) is well known as an anti-inflammatory and antiplatelet agent. It is regularly used for secondary prophylaxis in myocardial infarction and ischemic apoplexy and for preventing arterial thrombosis after vascular surgery. Furthermore, an anti-carcinogenic effect of ASA has been discussed since the late 1980s/early 1990s. For the first time, a reduction in colon cancer incidence was observed by Kune et al. in a case–control study in the population of Melbourne (Australia) after regular ASA intake. When 715 patients with colorectal carcinoma and 727 control patients were interviewed, there was a significant difference with respect to ASA intake in favor of the control group [12].

Numerous other case–control studies have confirmed the preventive effect of NSAIDs in general or of ASA concerning colorectal cancer incidence [13,14,15]. Among others, Friis et al. reported a 27% risk reduction of colorectal cancer by ASA low-dose therapy for at least five years [16]. The anti-carcinogenic effect was also shown in a prospective study. Here, the colon cancer death rate was reduced after ASA therapy in patients not previously affected by the disease at the start of the study [17]. ASA also shows a preventive effect concerning the development of carcinomas of various entities, such as hepatocellular carcinomas [18], pancreatic carcinomas [19], breast cancer [20], as well as numerous other entities [21].

In addition, individual studies showed a therapeutic ASA effect on tumors. For example, in a prospective study of patients with colorectal carcinoma, a lower mortality was observed after ASA therapy [22]. Similarly, lower mortality in patients with breast cancer treated with ASA was demonstrated in an observational study [23].

Numerous papers deal with the anti-carcinogenic effect of ASA, but the exact mechanism is still unclear. A promising approach to elucidate the mechanism of action of the anti-carcinogenic effect of ASA relates to tumor angiogenesis [24,25,26]. ASA acts on tumor vessels in two ways. On the one hand, it normalizes the disturbed tumor vessel architecture; on the other hand, it inhibits neo-angiogenesis [24].

The role of tumor angiogenesis has generally been in focus since 1971, when Folkman described its importance in terms of tumor growth and metastasis [27]. The vessels in tumors differ markedly from physiological vessels, caused by the unphysiologically rapid growth of tumors, dysregulation of angiogenic factors, inflammatory processes as part of the immune response, and mechanical or metabolic stress. Therefore, the arrangement of tumor vessels is disorganized, their diameter varies greatly, and shunts are common [28].

The composition of the vessel walls is also highly non-physiological. Some vessels are lined with tumor cells instead of endothelial cells; moreover, the endothelial cells can be malformed and mis-layered (grow on top of each other, and have less stable cell contacts. The consequence is increased vascular permeability [29,30]. The undirected vessel growth and the high interstitial pressure due to rapid tumor cell proliferation lead to vascular collapses, an inhomogeneous oxygen distribution, and areas with lower pH values [31]. This, in turn, leads to an increased release of angiogenic factors, which further intensifies excessive, unphysiological vascular growth [28].

At the molecular level, the main mediator of both physiological and tumor angiogenesis is the signaling molecules of the VEGF family and their receptors. VEGF is secreted mainly by tumor cells and numerous other cells, such as pericytes, macrophages, leukocytes, and platelets. Different VEGF subtypes are distinguished, of which VEGF-A is most involved in angiogenesis [32]. Via the tyrosine kinase receptors VEGFR-1, -2, and -3, intracellular signaling pathways are initiated via binding of VEGF. Among those receptors, VEGFR-2 is known to most-strongly trigger angiogenesis and vascular sprouting [33].

VEGF seems to play a role in the anti-angiogenic effect of ASA. For example, in a mouse sarcoma model, a reduction of VEGF-A and -C was demonstrated by immunohistochemistry after 14 days of ASA therapy [26]. Furthermore, Maity et al. described a reduction in VEGF expression under ASA therapy in an in vitro breast cancer model [24].

Although there is strong evidence that the anti-angiogenic effect of ASA is related to VEGF expression, no studies have been dedicated to the role of the VEGF receptors.

Thus, we investigated the effects of ASA on the regulation of VEGFR-2 in a murine breast cancer model. For this purpose, in addition to histological analysis, we used molecular contrast-enhanced ultrasound (CEUS) imaging to visualize the effects longitudinally, in vivo. To further clarify the anticarcinogenic effect of ASA, we also investigated the influence of ASA on tumor growth and perfusion, as well as vascular density and maturation.

## 2. Results

### 2.1. Tumor Growth

The manual measurement by caliper showed that tumors continuously grew in both groups. ASA treatment did not significantly affect tumor growth over the entire observation period. In the therapy group, the mean tumor volume on Day 1 (d1) was 21.94 ± 11.73 mm^3^. It had increased to 176.25 ± 50.50 mm^3^ on Day 7 (d7) and reached 345.06 ± 140.55 mm^3^ on Day 11.

In the control group, the initial tumor volume was 22.42 ± 13.5 mm^3^. At d7 and d11, it had increased to 150.7 ± 34.57 mm^3^ and 376.26 ± 73.06 mm^3^, respectively (Figure 1A).

In addition to manual measurement, volume measurement was performed on the ultrasound images. According to the manual measurement, there was continuous tumor growth in the therapy group. In the therapy group, the mean tumor volume on d1 was 15.45 ± 7.51 mm^3^. Towards Time Point d7, the volume increased to 132.09 ± 50 mm^3^. At Time Point d11, it was 286.26 ± 176.40 mm^3^. In the control group at d1, the volume was 18.58 ± 8.42 mm^3^ and increased to 115.83 ± 35.52 mm^3^ towards Time Point d7. At d11, there was a further increase up to 250.44 ± 74.18 mm^3^. In line with the results of the caliper measurement, the ultrasound volume measurement showed no significant difference be-tween the groups.

### 2.2. Relative Blood Volume

The relative intratumoral blood volume (rBV) was determined using contrast-enhanced ultrasound examinations (CEUS) with untargeted microbubbles. Here, the peak change in signal intensity in the tumor after injecting microbubbles served as a measure of tumor perfusion. No significant influence of ASA on rBV could be detected. In detail, the ASA-treated group showed a slight decrease in rBV from d1 to d7 (from 1.1 ± 0.73 arbitrary units (au) to 0.97 ± 0.25 au). Towards d11, there was a further decrease in rBV (0.75 ± 0.31 au).

The rBV of the control group started at 1.43 ± 0.75 au. Here, towards d7, the decrease in rBV was comparable to the ASA-treated group (1.19 ± 0.95 au). At d11, there was a further decrease in rBV to 0.61 ± 0.25 au. At none of the time points was there a significant difference between the groups (Figure 1B).

### 2.3. Molecular Ultrasound

The change of intratumoral VEGFR-2 expression over time was determined using molecular ultrasound imaging with VEGFR-2-targeted microbubbles.

On d1, destruction replenishment analyses revealed values of 0.30 ± 0.34 au for bound VEGFR-2-specific microbubbles in the therapy group. At d7, there was a marked increase in VEGFR-2-specific microbubble binding (1.31 ± 1.24 au). In contrast, at d11, the binding of VEGFR-2-specific microbubbles was decreased to 0.54 ± 0.55 au under ASA therapy.

The initial signal intensity values in the control group were 0.71 ± 0.75 au for bound VEGFR-2-specific microbubbles. Comparable to the therapy group, there was an initial increase in VEGFR-2 microbubble binding from d1 to d7 with signal intensity values of 1.32 ± 0.34 au at d7. However, unlike the therapy group, there was a further increase in VEGFR-2-specific microbubble binding towards d11 (2.24 ± 0.46 au), resulting in a significant difference from the therapy group (*p* = 0.0015) (Figure 2A,B).

### 2.4. Histological Analyses

Vessel densities were determined by quantification of CD 31 stainings. In addition, the VEGFR-2-positive area and the VEGFR-2-positive vessel fraction were calculated to assess the angiogenic status of the vasculature. At d1, the percentage of CD31-positive area fractions was 2.51 ± 0.21% on average. At d7, the CD31-positive area fraction in the therapy group was slightly lower (2.41 ± 0.37%). Toward d11, the CD31 positive area fraction increased to (2.89 ± 1.41%) under therapy.

In the control group, the area fraction was slightly higher at d7 (2.72 ± 0.63%). At d11, the percentage of CD31-positive area in the control group was higher than in the therapy group (3.46 ± 0.31%). Although there was no significant difference between the groups at d11 (*p* = 0.47), there was a trend towards lower vessel densities in ASA-treated tumors (Figure 3A,B).

The immunohistochemical evaluation of VEGFR-2 expression was consistent with the molecular ultrasound results. The VEGFR-2-positive area fraction at d1 was 0.35 ± 0.003% and slightly increased to 0.39 ± 0.12% in the therapy group at d7. Then, at d11, it dropped to 0.29 ± 0.26%.

In the control group, the value at d7 was also higher than at d1 (0.45 ± 0.35%). At d11, there was still a markedly higher VEGFR-2 area (0.9 ± 0.27%) and, thus, a significant difference from the therapy group (*p* = 0.002).

Furthermore, the ratio of VEGFR-2-positive vessel area confirmed the results of molecular ultrasound. At d1, the VEGFR-2-positive vessel fraction was 12.78 ± 1%. In line with the results from molecular ultrasound, the VEGFR-2-positive vessel area in the therapy group was higher at d7 (16.15 ± 4.37%). On d11, the histological analyses also confirmed the results from molecular ultrasound, showing a drop in the VEGFR-2-positive vessel fraction to 11.41 ± 0.93%.

In the control group, the VEGFR-2-positive area fraction was also higher at d7 (16.2 ± 0.56%) than at d1 (12.78 ± 1%). At d11, an even higher value was measured in the control group (26.18 ± 2.59%). Thus, according to the results of the molecular ultrasound, there was a significant difference between the therapy and control group (*p* = 0.00012) (Figure 3A,C). The fact that the VEGFR-2-positive vessel fraction changed indicated that there was a true change in the angiogenic status of the vessels and not just an overall change in vessel densities (that would also go along with a change in the VEGFR-2-positive image fraction).

To assess the effects on vessel maturation, α-smooth muscle actin (α-SMA) staining was performed, and the α-SMA-positive vessel fraction was calculated. At d1, the α-SMA-positive vessel fraction was 27.61 ± 4.77%. It constantly increased in the therapy group until d11 (d7: 47.21 ± 8.14%; d11; 49.08 ± 5.96%).

In the control group, the percentage also increased until d7 (51.26 ± 1.61%), but then, dropped markedly at d11 to 38.92 ± 2.08%. Thus, there was a significant difference at d11 between the therapy and control group for vascular maturity (*p* = 0.024) (Figure 4).

## 3. Discussion

This study explored the anti-cancerogenic effect of ASA by molecular ultrasound imaging. For this purpose, tumor volume and blood flow changes were observed in a murine breast cancer model over time. Furthermore, the impact of ASA on the VEGFR-2 expression on tumor vessels and on the vessel maturation was investigated histologically.

Within the observation period, ASA had no effect on tumor growth. Furthermore, no therapeutic effect could be detected concerning tumor perfusion assessed by CEUS. However, a significant effect on tumor angiogenesis could be detected by molecular ultrasound, represented by a significant decrease in the signal of VEGFR-2-bound microbubbles at d11. The effect on VEGFR-2 expression was confirmed by immunofluorescence microscopy. Furthermore, a tendency towards lower vessel densities and a significantly higher proportion of mature vessels under ASA therapy was observed histologically.

Regarding the tumor volume, an effect of ASA was not necessarily expected. The influence of ASA on tumor growth in murine models has been inconsistently described in the literature. For example, Miao et al. reported no effect on tumor growth by ASA therapy in a model of liver carcinoma [34], and Thakkar et al. showed no effect in a model of pancreatic cancer [35]. In contrast, Zhang et al. described a reduction in tumor growth in a murine sarcoma model [26]. However, it should be considered that a slightly lower dose was chosen in the former two studies and a slightly higher dose of 50 mg/kg in the latter. Given these data, the impression of a dose-dependent effect can be gained. However, it should be noted that, besides the dosing, the ASA responses may also strongly vary among different tumor models, so the results are not necessarily comparable. In the present study, low-dose therapy was chosen because the anti-carcinogenic effect of ASA was observed mainly with long-term low-dose therapy [16,36]. In addition, with regard to a possible clinical relevance, a significant increase in side effects at higher doses should be considered [37,38].

The focus of this work was less on assessing ASA’s effects on tumor growth, but on the tumor vessels. Several studies have shown that the anti-cancerogenic effect of ASA can be explained, among other mechanisms, by an anti-angiogenic effect, in particular by inhibiting VEGF [24,26]. An investigation of a possible effect on the corresponding receptor has not yet been performed. Thus, our study demonstrated an inhibitory effect on VEGFR-2 for the first time.

Inhibition of excessive angiogenesis in the tumor may lead to lower vessel densities and more physiological vessel growth. These considerations are consistent with observations of other studies showing vessel normalization under VEGF inhibition [39].

In the present study, the observed inhibition of VEGFR-2 expression resulted in a tendency towards a reduced vessel density, as indicated by CD31 staining. This reduction in CD31 is in line with the current state of research. For example, Huang et al. showed a significant reduction of CD31 under ASA in an ovarian cancer model over a longer observation period of 3–4 weeks [25].

However, functional CEUS showed no effect of ASA on the relative intratumoral blood volume, which does not necessarily have to be a contradiction to the reduced vessel density assessed by histological analyses. Angiogenesis in the tumor is highly non-physiological and chaotic, so the tumor vessels are not uniformly perfused and contain a high amount of non-perfused vessels [28]. This leads to an uneven supply of oxygen, which in turn leads to an excessive activation of pro-angiogenic mechanisms that increase unphysiological vessel growth [28]. Therefore, it can be concluded that a higher vessel density in the tumor, assessed histologically, does not necessarily have to be accompanied by better tumor perfusion.

Instead of blocking angiogenesis and, thus, cutting off the tumor from nutrient and oxygen supply, some anti-angiogenic therapy approaches aim to induce normalization of the intratumoral vasculature. Due to a more physiological vessel growth, tumor perfusion is improved, tumor cell migration is reduced, and there is a better precondition for transporting other therapeutic agents into the tumor [40,41,42]. The results of α-SMA staining showed this effect in the present work under therapy with ASA. The therapy led to a more physiological vessel growth, with a significantly higher proportion of mature vessels.

Another reason why no changes at the functional level were observed could be the relatively short observation period. This hypothesis is supported by the fact that the effects on VEGFR-2 could only be detected at the last observation time point. Further studies performed over a longer observation period, possibly on a slower-growing tumor model, could here provide more insights.

Further studies are also needed to elucidate the mechanism of action of ASA on VEGFR-2 expression. The most likely mechanism seems to be a COX-2-dependent effect, since the anti-inflammatory action, as the main pharmacological effect of ASA, also occurs via inhibition of COX-2. Other studies showed a correlation between COX-2 expression and tumor VEGF concentration [40,41], suggesting a COX-2-dependent reduction in VEGFR-2 expression. Another mechanism leading to a reduced VEGFR-2 expression could be the inhibition of FGF via COX-2 [42,43], as decreased FGF expression was shown to decrease VEGFR-2 expression [44].

It should also be considered that the anti-cancerogenic effect of ASA is not exclusively due to the inhibition of angiogenesis. Several other targets of ASA in tumors and its microenvironment have been described. Among others, Sharma et al. reported an immunological effect via dendritic cells through COX-2 inhibition [45]. Hsieh et al. described the influence of ASA on macrophage polarization in the tumor microenvironment [46]. Other studies described an anti-cancerogenic effect through COX-dependent platelet inhibition by ASA [47]. Again, further studies are needed to clarify the role of the different mechanisms.

In conclusion, ASA inhibited VEGFR-2 expression in a murine cancer model, which could lead to more physiological vessel growth and improved tumor perfusion. This strengthens previous observations and assumptions that ASA may have an anti-angiogenic tumor effect. Further mechanistic studies and longitudinal analyses on various tumor models are required to evaluate whether ASA could represent a valuable co-medication to inhibit cancer progression.

## 4. Materials and Methods

### 4.1. Animal Experiments

Animal experimentation was approved by the regulatory agency of North Rhine-Westphalia (Authority for Nature, Environment and Consumer Protection). Orthotopic murine 4T1 breast cancers were induced in immunocompetent, female BALB/cAnNRj mice (age: 6–8 weeks, Janvier Labs, Saint Berthevin, France). Mice were housed in groups of 3–5 animals on spruce granulate bedding (Lignocel, JRS, Rosenberg, Germany) under specific pathogen-free conditions in Type II long individually ventilated cages (Tecniplast, Hohenpeißenberg, Germany) with a 12 h light and dark cycle in a temperature- (20–24 °C) and humidity-controlled (45–65%) environment according to the guidelines of the “Federation for Laboratory Animal Science Associations” (FELASA, www.felasa.eu, accessed on 4 December 2017). One nestlet per cage was provided to enable nest building. Water and standard pellets for laboratory mice (Sniff GmbH, Soest, Germany) were offered ad libitum. Group-housed animals were assigned to individual earmarks for identification. A total of 19 mice were injected with 40,000 tumor cells in 50 µL PBS, under inhalation anesthesia (isoflurane 2%, 98% oxygen), in the area of a right mammary fat pad.

Tumor size was assessed daily via caliper measurements. Tumor volumes were calculated using the formula 0.52 × tumor length × tumor width^2^. The start of therapy (d1) was determined when the largest tumor extension of 3–4 mm was reached. The mice were randomized into the following groups: therapy ASA and control (*n* = 5 mice per group). Histological evaluation was performed on Day 1 on three additional mice and on Day 7 on three additional mice per group.

In the therapy group, mice received a daily intraperitoneal injection of 40 mg/kg body weight ASA (Aspirin i.v., Bayer AG, Leverkusen, Germany) dissolved in 300 μL PBS starting at d1. The dosage of ASA was chosen as a low-dose therapy based on the research findings of Stark et al. [48]. In the control group, 300 μL PBS was injected intraperitoneally daily.

### 4.2. Functional Contrast-Enhanced Ultrasound 

CEUS was performed at therapy Day 1 (d1), Day 7 (d7), and Day 11 (d11) using the VEVO 2100 ultrasound device (FUJIFILM-VisualSonics, Toronto, ON, Canada) and the 18 MHz transducer MS-250. The non-linear contrast mode was applied with a mechanical index of 0.03. The focus was placed on the tumor center. After the tumor was located by ultrasound, 1 × 10^8^ non-targeted (poly-(butyl cyanoacrylate) microbubbles (SonoMAC-r; SonoMAC, Aachen, Germany) in 50 µL 0.9% NaCl were injected via a tail vein. The baseline of signal intensity and inflow of the contrast medium in the tumor was recorded over a time of 25.5 s from injection with a frame rate of 10 frames per second. Signal intensities on ultrasound images were determined using the Imalytics Preclinical software (Gremse-IT GmbH, Aachen, Germany). The difference between the signal intensity before injection of microbubbles and the peak value after injection was determined to assess the relative blood volume.

### 4.3. Molecular Ultrasound Imaging

Streptavidin-coated (poly-(butyl cyanoacrylate) microbubbles (SonoMAC-t; SonoMAC, Aachen, Germany) were decorated with biotinylated VEGFR-2 antibodies (eBiosciense, Thermo Fisher Scientific Inc., Waltham, MA, USA). Then, 5 × 10^7^ VEGRF-2-specific microbubbles were injected in 50 µL 0.9% NaCl via a tail vein. The inflow of VEGFR-2-specific microbubbles was recorded using the same settings as for the functional CEUS. Eight minutes after injection, when most unbound microbubbles were cleared from the blood, the signal of receptor-bound microbubbles was determined using the destruction–replenishment method, as described previously by Byzl et al. [49] (for a more detailed explanation of the destruction-replenishment method, please refer to Appendix A).

Since VEGFR-2-specific binding of microbubbles has already been demonstrated in previous work, the use of control microbubbles was omitted [10,50].

### 4.4. Ultrasound Volume Measurement

In addition to manual measurement, tumor volume measurements were also performed based on B-mode ultrasound images. The length and width of the tumors were measured, and according to the formula 0.52 × tumor length × tumor width^2^, the volume was calculated.

### 4.5. Immunofluorescence Analysis

For histological analyses, three animals were euthanized on Day 1, six on Day 7 (three per group), and ten on Day 11 (five per group). All tumors were excised and cryopreserved in Tissue-Tek (Sakura, Zoeterwoude, The Netherlands) for histologic analysis. Frozen tumors were cut into 8 µm-thick slices. Slices from each at the largest diameter were used for VEGFR-2 staining and α-smooth muscle actin (α-SMA) staining. A rat anti-mouse CD31 antibody (BD Biosciences, Heidelberg, Germany) was used for vessel staining. VEGFR-2 was stained using a goat anti-mouse VEGFR2-antibody (R&D Systems, Wiesbaden, Germany), and α-SMA was stained using a biotinylated anti-α-smooth muscle actin (α-SMA) antibody (Progen, Heidelberg, Germany). As secondary antibodies, anti-rat AlexaFluor 488, anti-goat Cy-3, and streptavidin-Cy-3 were used (all from Dianova, Hamburg, Germany), respectively. Cell nuclei were counterstained by 4′,6-diamidino-2-phenylindole (Invitrogen, Thermo Fisher Scientific Inc., Waltham, MA, USA).

The stained sections were evaluated under an epifluorescence microscope (Axio Imager.M2, Carl Zeiss Microimaging, Göttingen, Germany) with a high-resolution camera (AxioCam MRm Rev.3, Carl Zeiss Microimaging, Göttingen, Germany). After examination of the entire tumor slice, five representative image sections were photographed for further evaluation. Quantification of CD31 and VEGFR-2 positive areas was performed using ImageJ 1.50i [51]. Finally, the percentage of the VEGFR-2-positive on the CD31-positive area fraction was calculated.

For the evaluation of α-SMA staining, CD31-positive and α-SMA-positive vessels were counted manually. The percentage of α-SMA-positive vessels to CD31-positive vessels was determined.

### 4.6. Statistical Analysis

Statistical analysis was performed using Microsoft Excel (Microsoft Corporation, Redmond, WA, USA). The results are presented as the means ± the standard deviations. Due to the different variances of the different groups, statistical significance was evaluated using Welch’s *t*-test. Results corresponding to *p* ≤ 0.05 were considered statistically significant.

## Figures and Tables

**Figure 1 ijms-24-07060-f001:**
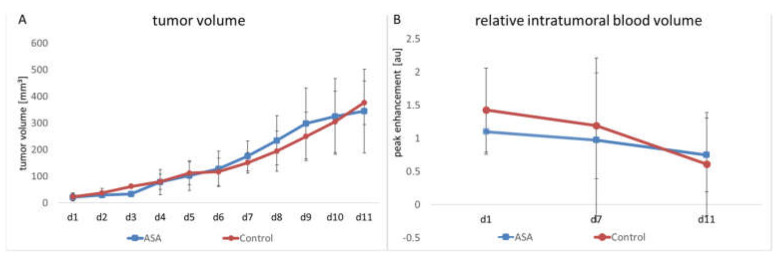
Changes in tumor volume and relative intratumoral blood volume in ASA-treated and -untreated tumors. The therapy and control groups do not significantly differ in tumor volume (**A**) and rBV (**B**), the latter measured by CEUS (indicated are the means ± SD) from day 1 to day 11 (d1–d11).

**Figure 2 ijms-24-07060-f002:**
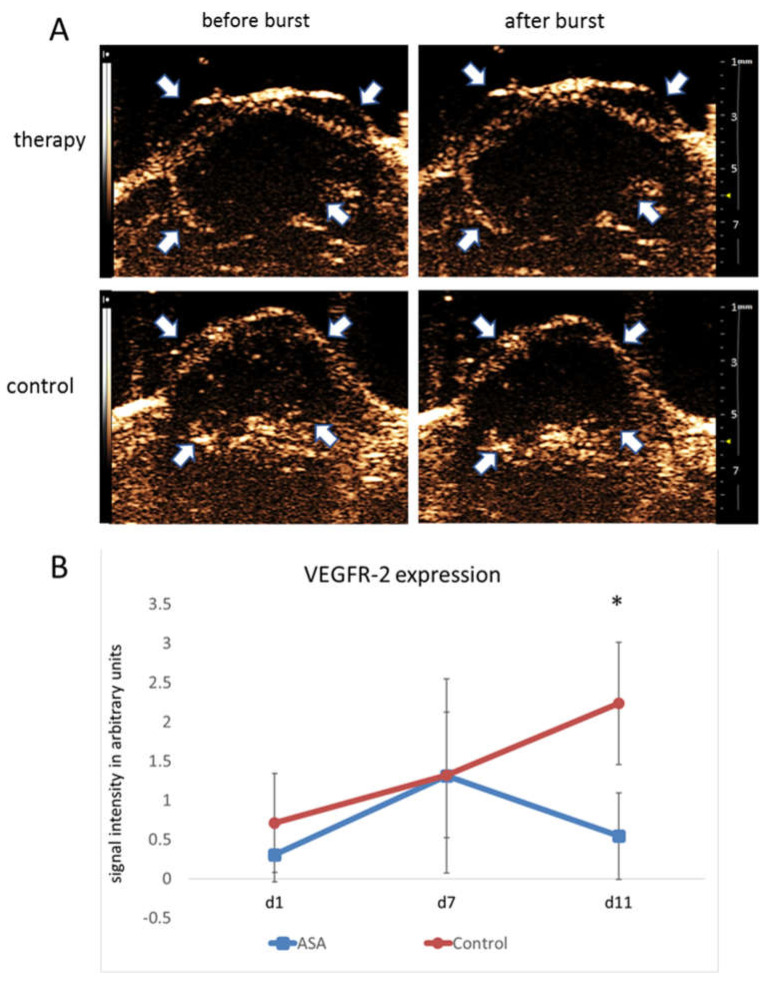
Molecular ultrasound imaging. (**A**) Representative ultrasound images of 4T1 tumors (contrast mode) after injection of VEGFR-2-targeted microbubbles indicate significantly lower binding in ASA-treated than -untreated tumors at d11. Arrows point to the tumor margins. (**B**) Quantitative analysis of signal intensities of bound VEGFR-2-specific microbubbles confirm the visual impression (indicated are the means ± SD; *: *p* < 0.05).

**Figure 3 ijms-24-07060-f003:**
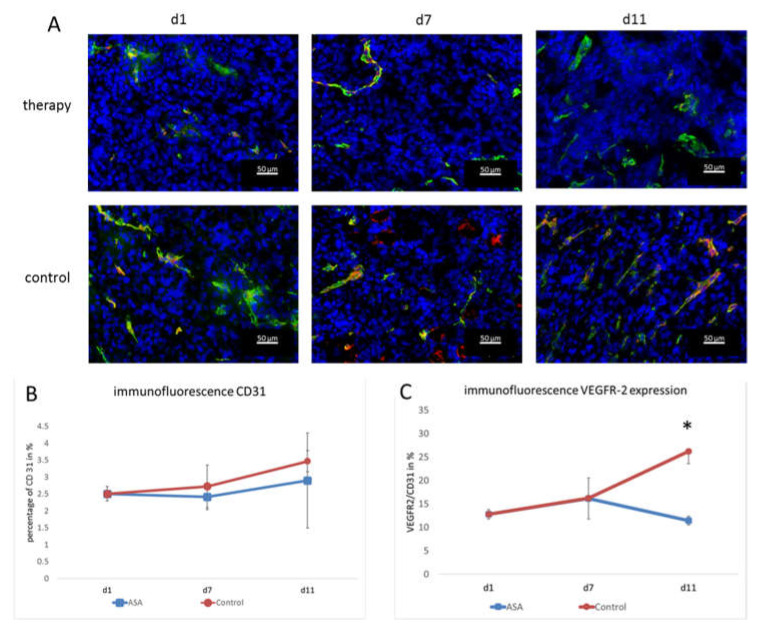
Immunofluorescence analysis and quantification of VEGFR-2 and CD31. (**A**) Representative immunofluorescence images showing CD31 (green), VEGFR-2 (red), and DAPI (blue). (**B**) The quantification of the CD31-positive area fraction (mean in % ± SD) shows a tendency towards lower values in the therapy group. (**C**) Fraction of CD31-positive vessels that co-localize with VEGFR-2 immunostaining (means ± SD, *: *p* < 0.05).

**Figure 4 ijms-24-07060-f004:**
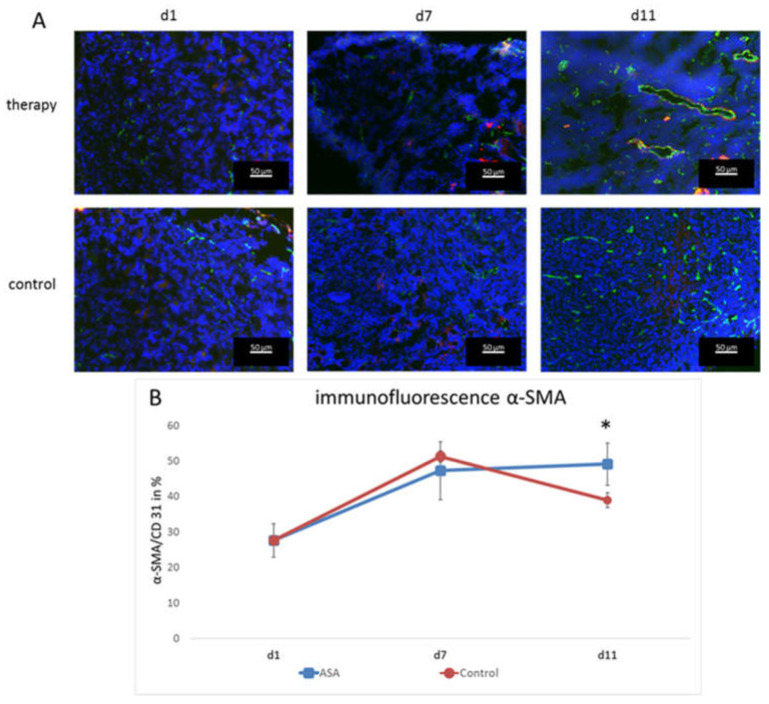
Immunofluorescence analysis and quantification of α-SMA-positive vessels. (**A**) Representative immunofluorescence images showing CD31 (green), α-SMA (red), and DAPI (blue). (**B**) Fraction of CD31-positive vessels that co-localize with α-SMA immunostaining (means ± SD, *: *p* < 0.05).

## Data Availability

The methods and materials used are fully described in the manuscript, and the data are available from the corresponding authors upon request.

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
