# Peer review of "Molecular Ultrasound Imaging Depicts the Modulation of Tumor Angiogenesis by Acetylsalicylic Acid"

_ijms, 2023, doi:10.3390/ijms24087060_

Round 1
Reviewer 1 Report
In this paper, authors applied VEGFR-2-targeted molecular ultrasound to explore a potential inhibitory effect of acetylsalicylic acid (ASA) on tumor angiogenesis in vivo by using the syngeneic mouse model of 4T1 mouse mammary tumors. They measured intratumoral blood volume (rBV) and the binding of VEGFR-2-specific microbubbles at different time points in control and mice treated with ASA 40mg/kg/day. A tendency towards reduced VEGFR-2-microbubbles in ASA treated tumors, while no difference in tumor volume and in CD31 positive cells. A confirmation of VEGFR2 reduction in CD31 positive cells was performed by IF and quantification. A total of 3 Figures containing 2-3 panels is presented and the main conclusion is that inhibition of angiogenesis via VEGFR-2 downregulation as one of the anti-tumor effects of ASA.
Major comments:
The relational of the study is not clear to the reviewer as whether authors try to verify the hypothesis of potential antitumor and antiangiogenetic effect of ASA or to present a technical paper that use VEGFR2 micro-bubbles for ultrasound quantification of tumor angiogenesis in live tumors. Clarifications are needed.
If there no difference in CD31 positive cells and no difference in tumor volume, how VEGFR2 downregulation is relevant to assess tumor angiogenesis?
Several bias can explain the reduction of VEGFR staining. Either ASA interferes with VEGFR2-micro-bubbles and blocks the binding of the anti-VEGFR2 antibody.
If there is no difference in tumor volume, authors should measure tumor weight at the day of sacrifice.
What is the effect of higher doses of ASA?
what is the control for VEGFR-micro bubbles? free bubbles?
What the arrows in figure 2A show?
Reviewer 2 Report
This very short article uses so-called molecular ultrasound as well as conventional immunohistochemistry (IHC) to show that in murine mammary tumours the binding of VEGFR-targeted microbubbles appears reduced by ASA treatment, and this observation correlates with a reduced expression of VEGFR2 in CD31+ BVs as determined by IHC. Technically this is interesting, but I would have liked to see a longer window of observation and possibly in several tumour types before making any firm conclusions regarding the mechanism of ASA.
Other points
1. The last paragraph appears to contain contradictions? 'Improved tumor perfusion' 'may have antiangiogenic tumor effect'
2. What was the area of sections scanned by Image J?
3. Fig. 2: indicate what arrows denote (what are we looking at?), and what is 'before' and 'after burst' mean, is this after microbubble injection?
4. Fig 3 legend, C: would it be clearer to say 'Fraction of CD31-positive vessels that co-localise with VEGFR2 immunostaining'
5. Typographical errors: Fig. 1 legend, 'later' should be 'latter'; line 131 'there was' should be 'was there'
Round 2
Reviewer 1 Report
Authors have addressed my comments and I have no further questions.